# Hypothalamic supramammillary neurons that project to the medial septum modulate wakefulness in mice

Mengru Liang[1,8], Tingliang Jian[2,3,8], Jie Tao[4,8], Xia Wang[5], Rui Wang[2], Wenjun Jin[2], Qianwei Chen[2], Jiwei Yao[5], Zhikai Zhao[5], Xinyu Yang[5], Jingyu Xiao[6], Zhiqi Yang[2], Xiang Liao [5], Xiaowei Chen [2,7✉], Liecheng Wang [1✉] & Han Qin [5,7✉]

The hypothalamic supramammillary nucleus (SuM) plays a crucial role in controlling wakefulness, but the downstream target regions participating in this control process remain unknown. Here, using circuit-specific fiber photometry and single-neuron electrophysiology together with electroencephalogram, electromyogram and behavioral recordings, we find that approximately half of SuM neurons that project to the medial septum (MS) are wake-active. Optogenetic stimulation of axonal terminals of SuM-MS projection induces a rapid and reliable transition to wakefulness from non-rapid-eye movement or rapid-eye movement sleep, and chemogenetic activation of SuM$^{MS}$ projecting neurons significantly increases wakefulness time and prolongs latency to sleep. Consistently, chemogenetically inhibiting these neurons significantly reduces wakefulness time and latency to sleep. Therefore, these results identify the MS as a functional downstream target of SuM and provide evidence for the modulation of wakefulness by this hypothalamic-septal projection.

[1] Department of Anatomy, School of Basic Medical Sciences, Anhui Medical University, Hefei 230032, China. [2] Brain Research Center and State Key Laboratory of Trauma, Burns, and Combined Injury, Third Military Medical University, Chongqing 400038, China. [3] Farm Animal Genetic Resources Exploration and Innovation Key Laboratory of Sichuan Province, Sichuan Agricultural University, Chengdu 611130, China. [4] Advanced Institute for Brain and Intelligence, School of Medicine, Guangxi University, Nanning 530004, China. [5] Center for Neurointelligence, School of Medicine, Chongqing University, Chongqing 400044, China. [6] Department of Anesthesiology, Chongqing University Cancer Hospital, Chongqing 400030, China. [7] Chongqing Institute for Brain and Intelligence, Guangyang Bay Laboratory, Chongqing 400064, China. [8] These authors contributed equally: Mengru Liang, Tingliang Jian, Jie Tao.
✉email: xiaowei_chen@tmmu.edu.cn; wangliecheng@ahmu.edu.cn; qinhan@cqu.edu.cn

The supramammillary nucleus (SuM) is a hypothalamic region lying above the mammillary body and provides abundant projections to numerous brain regions like the hippocampus, septum, frontal cortex, and cingulate cortex[1,2]. Recent advances in high-performance recording and manipulation techniques have enabled extensive studies of SuM functions, subsequently revealing its involvement in numerous processes such as episodic memory[3,4], novelty detection[5], theta rhythm[6,7], locomotion[8], hippocampal neurogenesis[9], and wakefulness[10]. In particular, one previous study demonstrated that SuM glutamatergic neurons serve as a key node for arousal, and chemogenetic activation of SuM glutamatergic neurons, but not GABAergic neurons, produces sustained arousal[10]. However, which downstream brain regions are involved in the SuM control of arousal remains unknown.

The medial septum (MS), which primarily contains cholinergic, GABAergic and glutamatergic neurons[11,12], has been suggested to mediate different brain functions like locomotion[13], learning and memory[14,15], hippocampal theta generation[16], and wakefulness[17,18]. Among these functions, MS glutamatergic neurons were shown to control wakefulness by activating lateral hypothalamic glutamatergic neurons[18]. Furthermore, a recent study has demonstrated that SuM glutamatergic neurons project to MS glutamatergic neurons and are responsible for modulating the motivation for environmental interaction[19]. Based on this established anatomical connection and combined findings, we hypothesized that a SuM-MS projection may modulate wakefulness.

To test this hypothesis, we performed circuit-specific optical $Ca^{2+}$ and optrode recordings in SuM-MS projection across sleep-wakefulness cycles. We identified a set of wake-active neurons in SuM that projects to MS. Optogenetic or chemogenetic activation of SuM-MS projection induced behavioral and EEG arousal, and chemogenetic inhibition of this projection decreased wakefulness. Overall, our results reveal a critical role of the hypothalamic-septal projection for wakefulness modulation.

## Results

### SuM-MS projection terminals are strongly active during both wakefulness and REM sleep.

SuM neurons have been reported to project to MS region[1,20]. We labeled the SuM neurons by local injection of an adeno-associated viral (AAV) vector to deliver the enhanced green fluorescent protein (eGFP) gene into SuM (Supplementary Fig. 1a). Four weeks after injection, robust eGFP expression was observed in cell bodies within SuM (Supplementary Fig. 1b), and the axonal terminals in MS were also labeled with eGFP (Supplementary Fig. 1c). To further investigate the SuM to MS connection, a retrograde AAV vector expressing eGFP was injected into MS (Supplementary Fig. 1a). We verified that the expression area of eGFP was limited in MS (Supplementary Fig. 1d), and the corresponding eGFP-labeled cell bodies were observed in SuM (Supplementary Fig. 1e).

Previous studies have reported the presence of glutamatergic, GABAergic, and nitric oxide synthase (Nos1) positive neurons in SuM[5,7,10,21]. To determine whether the synaptic transmission of SuM-MS is excitatory or inhibitory, we injected AAV-Syn-ChR2-mCherry into SuM and recorded the postsynaptic currents in MS neurons while activating ChR2 with blue light pulses in brain slices (Supplementary Fig. 2a, b). Excitation of SuM resulted in large excitatory postsynaptic currents (EPSCs) with a latency of $5.3 \pm 0.6$ ms (Supplementary Fig. 2c–e, measured with the membrane held at $-70$ mV, $n = 14$ neurons). In contrast, no significant inhibitory postsynaptic currents (IPSCs) were observed (Supplementary Fig. 2c, d, measured at $+10$ mV, $n = 14$ neurons). Moreover, the light-induced EPSCs were

suppressed by the sodium channel blocker tetrodotoxin (TTX) and subsequently recovered upon application of the potassium channel blocker 4-Aminopyridine (4-AP; see example in Supplementary Fig. 2f; statistics in Supplementary Fig. 2g, ACSF: $n = 14$ neurons, TTX: $n = 4$ neurons, TTX & 4-AP: $n = 7$ neurons; also see statistics in Supplementary Data 1 for all figures). These findings indicate that SuM neurons provide monosynaptic excitatory connections to MS neurons.

Next, we investigated the Nos1 immunoreactivity in mice with mCherry-labeled SuM neurons projecting to MS (referred to as SuM^MS projecting neurons), achieved by injecting retroAAV-Cre into MS and AAV-DIO-mCherry into SuM. We found Nos1-containing neurons distributed in SuM as previously described[10,21], and approximately $17.6 \pm 1.6\%$ of SuM^MS projecting neurons expressing Nos1 (Supplementary Fig. 3, $n = 8$ mice).

Although both SuM and MS neurons have been shown to function as essential components in wakefulness[10,18], the activity has not been recorded during sleep-wakefulness cycles. First, a circuit-specific fiber photometry system[22–24] was used in conjunction with electroencephalogram (EEG) and electromyogram (EMG) recordings to observe $Ca^{2+}$ activities at axonal terminals of SuM-MS projection in freely moving mice. For this purpose, AAV-Syn-axon-jGCaMP7b[25,26] was locally injected into SuM to express the $Ca^{2+}$ indicator, jGCaMP7b, in axons of SuM neurons (Fig. 1a). One month following virus injection, an optical fiber was implanted with the tip above MS to record activity at axonal terminals of SuM neurons, and EEG-EMG electrodes were attached to the mouse cortical surface and neck muscles respectively, to define sleep-wakefulness states (Fig. 1a). Virus expression and fiber tip location were verified by post-hoc histology after recording finished (Fig. 1b, Supplementary Fig. 4). Notably, axonal terminals of SuM^MS projecting neurons had higher levels of $Ca^{2+}$ activity during both wakefulness and (rapid-eye movement) REM sleep than that during (non-rapid-eye movement) NREM sleep (Fig. 1c, d). In addition, these activities increased strongly during NREM-wakefulness and NREM-REM transitions, but sharply decreased during wakefulness-NREM transitions (Fig. 1e–h). Taken together, these results suggested that SuM projects to MS and the $Ca^{2+}$ activity in this projection is highly active during wakefulness and REM sleep.

### Identification of wake-active SuMMS projecting neurons.

To characterize the firing rates of SuM^MS projecting neurons at the single-cell level, we conducted optrode recordings across sleep-wakefulness cycles[27,28]. Channelrhodopsin-2 (ChR2) was expressed specifically in SuM^MS projecting neurons by injecting a Cre-dependent retrograde AAV (retroAAV-Cre) into MS and, concurrently, an AAV vector carrying DIO-ChR2-mCherry into SuM. We then implanted an optrode in SuM to identify SuM^MS projecting neurons and record single-neuron activities (Fig. 2a; see optrode locations in Supplementary Fig. 5). A series of blue light pulses (450 nm, 2 Hz, 10 mW, 10 ms duration) were delivered to stimulate ChR2-expressing neurons. SuM neurons were then identified as MS projecting neurons based on light-induced spikes which respond at short latency, low jitter, high success rate, and high correlation with spontaneous spike waveform (latency $3.6 \pm 0.3$ ms, jitter $0.8 \pm 0.1$ ms, success rate $95\% \pm 2\%$, correlation coefficient $0.92 \pm 0.02$, $n = 23$ neurons, Fig. 2b-e).

We found that two groups of neurons showed distinct firing features across sleep-wakefulness cycles. Neurons in one group significantly increased firing rates following the transition from NREM or REM sleep to wakefulness, and significantly decreased their firing rates following the switch from wakefulness to NREM sleep (wake-active neuron, see example in Fig. 2f). Neurons in the other group significantly increased firing rates following the

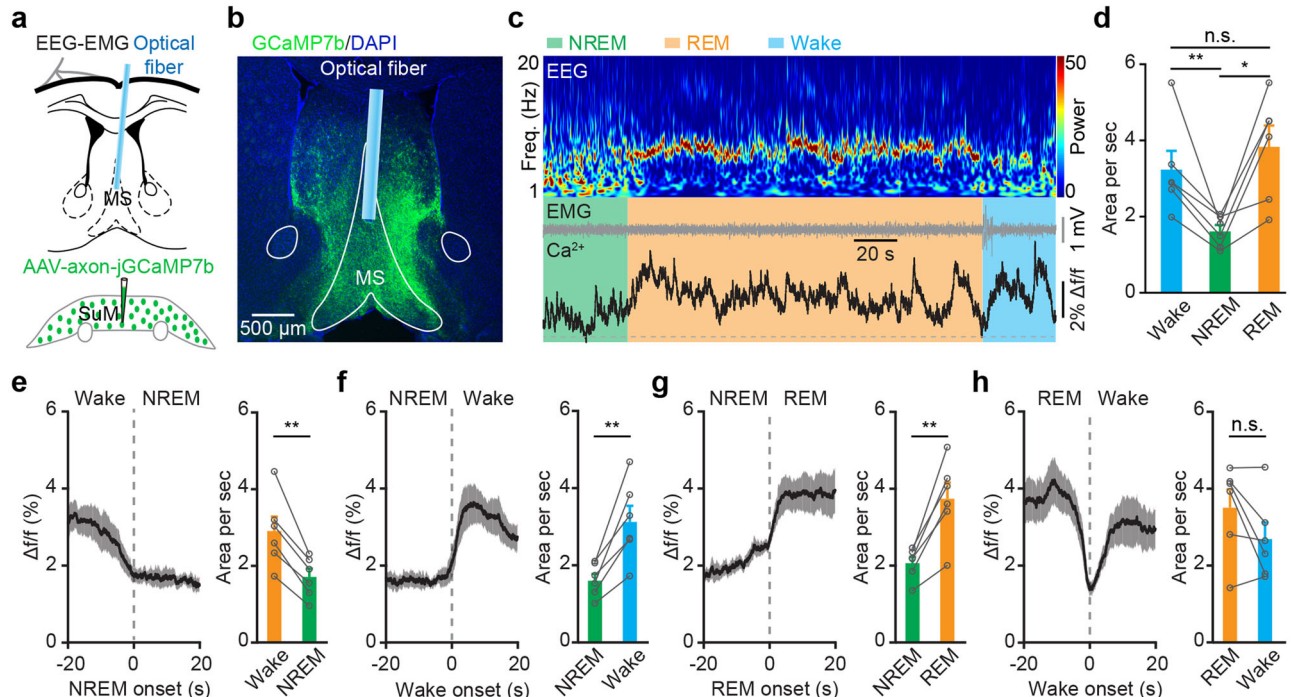

**Fig. 1 Strong activation of SuM-MS projection terminals during wakefulness and REM sleep. a** Experimental design for virus injection into SuM, fiber implantation in MS, and EEG-EMG recording. **b** Post-hoc histological images showing the expression of jGCaMP7b at axonal terminals of SuM neurons and optical fiber location in MS. **c** $Ca^{2+}$ activities in axonal terminals of SuM-MS projection across sleep-wakefulness cycles. Heatmap of EEG power spectrum ($\mu V^2$). Freq., frequency. **d** Summary of the area under the curve per second during wakefulness, NREM sleep, and REM sleep. $n = 6$ mice, RMs 1-way ANOVA with LSD post-hoc comparison, *$p < 0.05$, **$p < 0.01$. $Ca^{2+}$ activities during brain state transitions: wakefulness-NREM (**e**), NREM-wakefulness (**f**), NREM-REM (**g**), and REM-wakefulness (**h**). $n = 6$ mice, paired $t$-test, **$p < 0.01$, shading, SEM. Data are represented as mean ± SEM.

transition from NREM sleep to REM sleep and significantly decreased their firing rates following the switch from REM sleep to wakefulness (REM-active neuron, see example in Fig. 2g).

All recorded $SuM^{MS}$ projecting neurons were divided into wake-active (Fig. 2h, blue dots, $n = 13$ neurons) and REM-active neurons (Fig. 2h, black dots, $n = 10$ neurons) by analyzing the firing modulation based on the firing rates during wakefulness, NREM sleep and REM sleep states. We analyzed and compared the firing rates of wake-active and REM-active $SuM^{MS}$ projecting neurons across these three states. The firing rates of wake-active neurons during wakefulness were significantly higher than that during NREM or REM sleep (Fig. 2i left; wakefulness, $15.0 \pm 2.8$ Hz; NREM, $4.8 \pm 1.9$ Hz; REM, $6.6 \pm 2.4$ Hz; Friedman's ANOVA and Wilcoxon signed-rank tests, $n = 13$ neurons). Conversely, the firing rates of REM-active neurons during REM sleep were significantly higher than that during NREM sleep or wakefulness (Fig. 2i right; wakefulness, $9.0 \pm 2.9$ Hz; NREM, $8.9 \pm 3.0$ Hz; REM, $15.8 \pm 4.7$ Hz; Friedman's ANOVA and Wilcoxon signed-rank tests, $n = 10$ neurons). Furthermore, analysis of state transition revealed that these wake-active neurons increased their firing rates during NREM-wakefulness or REM-wakefulness transitions, but decreased their firing rates during wakefulness-NREM transitions (Fig. 2j, paired $t$-test). Burstiness analysis demonstrated that wake-active neurons exhibited a higher frequency of bursts than REM-active neurons (Supplementary Fig. 6c, d, Kolmogorov-Smirnov test, $P = 0.02$)[29,30]. These results established that wake-active neurons were indeed present in SuM-MS projection, likely contributing to the modulation of wakefulness.

**Stimulating SuM-MS projection promotes wakefulness.** To determine whether the SuM-MS projection indeed plays a key

role in the modulation of wakefulness, optogenetic activation was applied in MS to activate ChR2-expressing axonal terminals of SuM neurons (Fig. 3a). To this end, we injected AAV-ChR2-mCherry into SuM to express ChR2 in axonal terminals of SuM-MS projection (see virus expression in Fig. 3b, c and Supplementary Fig. 7a), and an optical fiber was subsequently implanted into MS to deliver blue light (see fiber location in Fig. 3c and Supplementary Fig. 7b, c). EEG-EMG electrodes were attached to monitor activity during sleep-wakefulness cycles, and axonal terminals of SuM-MS projection were optogenetically activated for 20 s (473 nm, 10 mW, 10 ms duration) after the onset of stable NREM or REM sleep. A control group of mice with activation of SuM to dentate gyrus (SuM-DG) projection, known to be associated with spatial memory and theta rhythm[4,5,7], was included (Supplementary Fig. 7d). The activation-induced transition from NREM sleep to wakefulness in a frequency-dependent manner (example in Fig. 3d; statistics in Fig. 3e, f). The success rate of transition from NREM sleep to wakefulness after 20-Hz optogenetic activation was 100%, with a latency to wakefulness of $2.0 \pm 0.3$ s (SuM-MS ChR2: $n = 10$ mice; SuM-DG ChR2 control: $65.8 \pm 11.8$ s, $n = 6$ mice; SuM-MS mCherry control: $65.1 \pm 6.0$ s, $n = 8$ mice). Transition to wakefulness was also induced upon 20-Hz optogenetic activation of this projection during REM sleep (example in Fig. 3g; statistics in Fig. 3h, i), with an $89\% \pm 8\%$ success rate and a latency to wakefulness of $16.4 \pm 6.7$ s (SuM-DG ChR2 control: $64.7 \pm 3.7$ s, SuM-MS mCherry control: $59.2 \pm 7.5$ s).

Previous studies have demonstrated the involvement of SuM in EEG activity[10,31]. We conducted an analysis of the EEG spectrum before and during light stimulations at different delivery frequencies. During NREM stimulation, activation of SuM-MS projection led to a decrease in power within the slow-wave and delta bands, while showing an increase in power within the

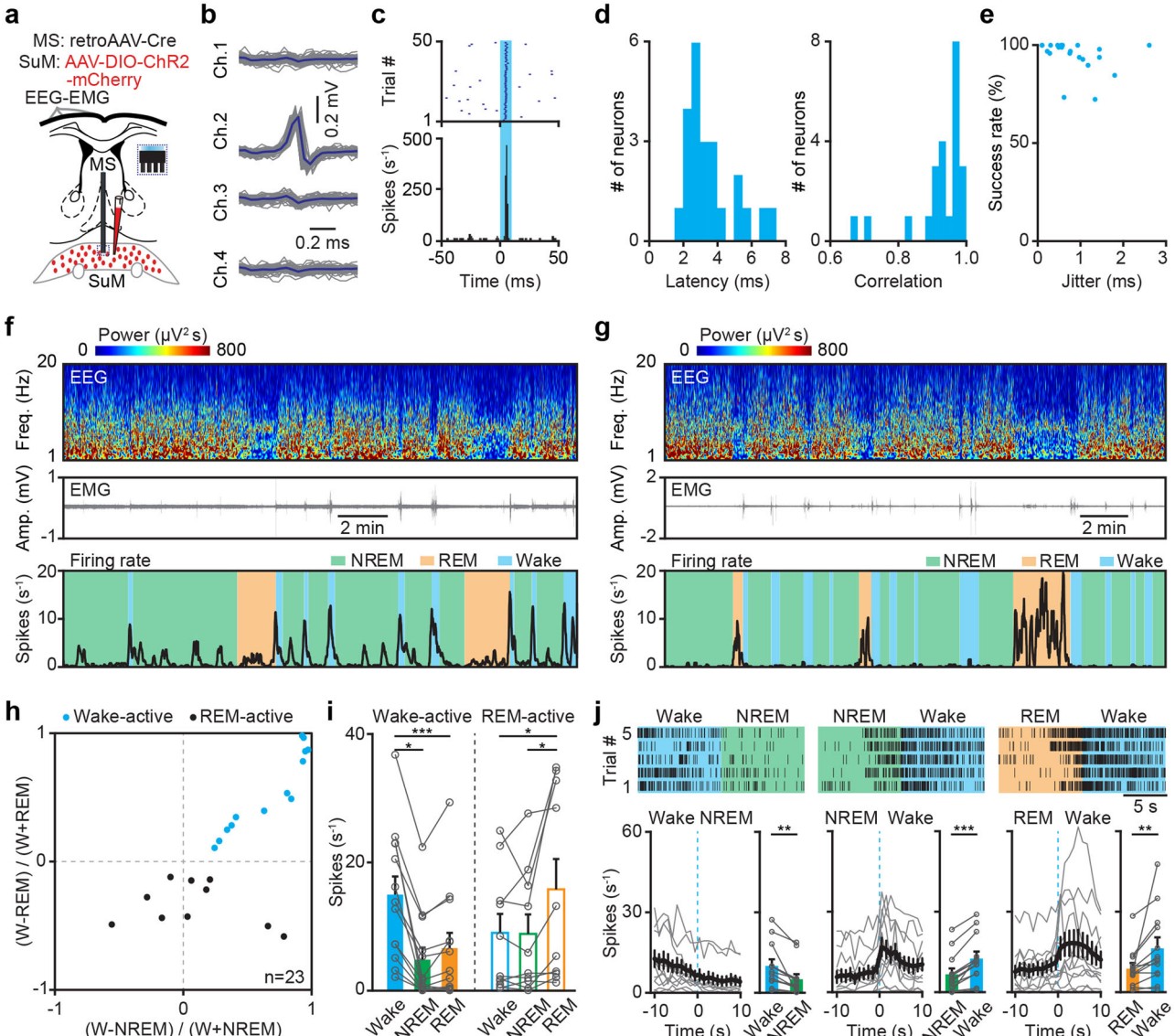

**Fig. 2 Optrode recording of wake-active SuM^MS projecting neurons. a** Experimental design for retrograde labeling of SuM^MS projecting neurons, optrode recording in SuM, and EEG-EMG recording. **b** Waveforms of average light-invoked (blue) and individual spontaneous (gray) spikes from a representative SuM^MS projecting neuron. **c** Stimulus time histogram of neuronal spikes in **b**. **d** Distributions of latencies of the first light-induced spikes (left), and correlation coefficients between light-induced spikes and spontaneous spikes (right) for all recorded SuM^MS projecting neurons, $n = 23$ neurons from 8 mice. **e** Success rate versus temporal jitter of the first light-induced spikes for all recorded SuM^MS projecting neurons, $n = 23$ neurons from 8 mice. **f** Firing rates of a representative wake-active neuron across sleep-wakefulness cycles. **g** Firing rates of a representative REM-active neuron across sleep-wakefulness cycles. **h** Firing rate modulation of SuM^MS projecting neurons, $n = 23$ neurons from 8 mice. **i** Summary of firing rates from wake-active (left, $n = 13$ neurons from 6 mice) and REM-active SuM^MS projecting neurons (right, $n = 10$ neurons from 5 mice) in different states. Friedman's ANOVA test with post hoc comparison, *$p < 0.05$, ***$p < 0.001$. **j** Firing rate of wake-active SuM^MS projecting neurons during state transitions: wakefulness-NREM, left; NREM-wakefulness, middle; REM-wakefulness, right. Top: example of a wake-active SuM^MS projecting neuron during different state transitions. Bottom left: average firing rates during state transitions. Bottom right: summary of firing rates of 10 s before and after state transitions, paired *t*-test, *$p < 0.05$, **$p < 0.01$, ***$p < 0.001$, $n = 13$ neurons. Data are represented as mean ± SEM.

low-theta (4–7 Hz), high-theta (7–12 Hz), and alpha bands (Fig. 3j). Moreover, during REM sleep, light stimulation resulted in a decrease in low-theta power and an increase in beta power during EEG spectral analysis before and during optogenetic activation (Fig. 3k, at 10 and 20 Hz).

To verify the above results, SuM^MS projecting neurons were selectively chemogenetically activated by specific labeling with an engineered G_i-coupled hM3Dq receptor[32]. We injected retroAAV-Cre into MS (see Supplementary Fig. 8a for the infection range) and AAV-DIO-hM3Dq-mCherry into SuM (Fig. 4a) to label these SuM^MS projecting neurons. Additionally,

the robust expression of hM3Dq-mCherry in SuM neurons was confirmed by post-hoc histological analysis (Fig. 4b). Immunostaining for c-Fos protein (a marker of active neurons)[33,34] showed that SuM^MS projecting neurons were activated after application of the synthetic ligand clozapine-N-oxide (CNO, 1 mg/Kg). In hM3Dq-positive neurons, c-Fos expression was significantly higher in the clozapine-n-oxide (CNO) application group than in saline-treated control animals (Fig. 4b, c, Wilcoxon signed-rank test, $P < 0.001$).

At behavioral level, intraperitoneal injection with CNO at the start of the light period (8:00 a.m.) resulted in significantly greater

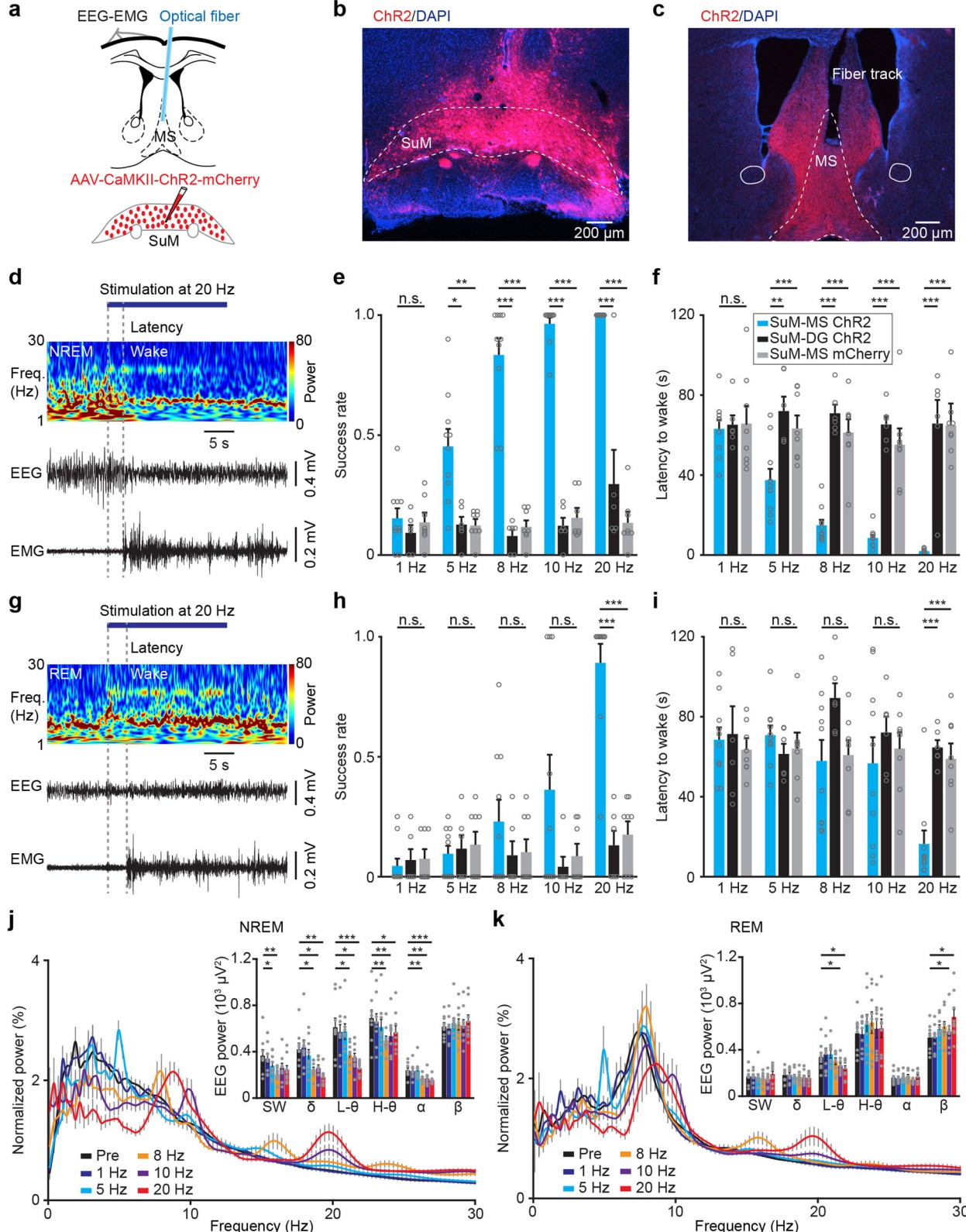

wakefulness (Fig. 4d–f; RMs 2-way ANOVA test, $P = 0.00006$; hM3Dq CNO and hM3Dq saline groups: $n = 8$ mice, mCherry group: $n = 7$ mice) and latency to first sleep was significantly longer in the hM3Dq CNO group (Fig. 4g; Kruskal-Wallis 1-way ANOVA test, $P = 0.0004$; hM3Dq CNO: $7.1 \pm 0.5$ h, $n = 8$ mice; hM3Dq saline: $0.7 \pm 0.1$ h, $n = 8$ mice and mCherry CNO: $0.5 \pm 0.2$ h, $n = 7$ mice) compared with that in the hM3Dq saline

and mCherry CNO control groups. In the hM3Dq CNO group, there was a significant increase in the average duration of wakefulness, while the duration of NREM and REM sleeps decreased (Fig. 4h, 1-way ANOVA test, $P < 0.01$). Furthermore, CNO treatment resulted in a significant increase in the high-theta band activity (1-way ANOVA test, $P = 0.006$), while there was a decrease in the low-theta band of EEG activity during

**Fig. 3 Optogenetic stimulation at axonal terminals of SuM neurons in MS induces wakefulness. a** Experimental design for virus injection into SuM, fiber implantation in MS, and EEG-EMG recording. Representative images showing mCherry-labeled somas in SuM (**b**), and mCherry-labeled axonal terminals of SuM neurons and the track of fiber in MS (**c**); Scale bar = 200 μm. **d** Representative EEG power spectrum ($\mu V^2$) and EEG-EMG trace data around 20-Hz stimulation during NREM sleep. Freq., frequency. **e** Summary of the success rate for inducing wakefulness from NREM sleep in different groups; SuM-MS ChR2, $n = 10$ mice, SuM-dentate gyrus (SuM-DG) ChR2, $n = 6$ mice, SuM-MS mCherry, $n = 8$ mice; RMs 2-way ANOVA with Sidak post-hoc comparison test, $*p < 0.05$, $**p < 0.01$, $***p < 0.001$. **f** Summary of the latency to wakefulness from NREM sleep; SuM-MS ChR2, $n = 10$ mice; SuM-DG ChR2, $n = 6$ mice; SuM-MS mCherry, $n = 8$ mice; RMs 2-way ANOVA with Sidak post-hoc comparison test, $**p < 0.01$, $***p < 0.001$. **g** Representative EEG power spectrum ($\mu V^2$) and EEG-EMG trace data around 20-Hz stimulation during REM sleep. **h** Summary of the success rate for inducing wakefulness from REM sleep; SuM-MS ChR2, $n = 10$ mice, SuM-DG ChR2, $n = 6$ mice, SuM-MS mCherry, $n = 8$ mice; RMs 2-way ANOVA with Sidak post-hoc comparison test, $***p < 0.001$. **i** Summary of the latency to wakefulness from REM sleep; SuM-MS ChR2, $n = 10$ mice; SuM-DG ChR2, $n = 6$ mice; SuM-MS mCherry, $n = 8$ mice; RMs 2-way ANOVA with Sidak post-hoc comparison test, $***p < 0.001$. **j** Spectral analysis of EEG activity before and during light stimulation in the NREM sleep state of SuM-MS ChR2 group; $n = 9$ mice, RMs 1-way ANOVA with LSD post-hoc comparison test, $*p < 0.05$, $**p < 0.01$, $***p < 0.001$. **k** Spectral analysis of EEG activity before and during light stimulation in the REM sleep state of SuM-MS ChR2 group; $n = 9$ mice, RMs 1-way ANOVA with LSD post-hoc comparison test, $*p < 0.05$. Data are represented as mean ± SEM.

wakefulness (1-way ANOVA test, $P = 0.01$). These experiments thus demonstrated that optogenetic and chemogenetic activation of SuM[MS] projecting neurons could promote wakefulness.

**Inhibition of SuM[MS] projecting neurons reduces wakefulness.** To further examine how chemogenetic inhibition of SuM[MS] projecting neurons affects the modulation of wakefulness, expression of an engineered $G_i$-coupled hM4Di receptor in SuM[MS] projecting neurons was induced by injecting retroAAV-Cre into MS and AAV-DIO-hM4Di-mCherry into SuM (Fig. 5a). Immunostaining detection of c-Fos verified that CNO injection led to the inhibition of SuM[MS] projecting neurons in hM4Di-expressing mice, indicated by lower c-Fos signal in the CNO group than in saline control (Fig. 5b, c, Wilcoxon signed-rank test, $P < 0.001$). Analysis of behavioral states (Fig. 5d) revealed a significant reduction in wakefulness duration during the first 2 hours following CNO injection in the hM4Di CNO group (Fig. 5e, RMs 2-way ANOVA test, $P = 0.003$). Moreover, the hM4Di CNO mice exhibited a shorter latency to first sleep compared to the control groups (Fig. 5f, 1-way ANOVA test, $P = 0.001$; hM4Di CNO: $14.3 \pm 2.0$ min, $n = 10$ mice; hM4Di saline: $26.6 \pm 3.3$ min, $n = 10$ mice; mCherry CNO: $30.4 \pm 3.0$ min, $n = 7$ mice). Further analysis revealed that the decrease of wakefulness in the hM4Di CNO group was primarily due to a reduction in the average episode duration of wakefulness rather than a change in the number of wakefulness episodes (Fig. 5g, h). Additionally, it was observed that optogenetic inhibition had no effect on EEG activity (Supplementary Fig. 9).

Likewise, the optogenetic inhibition of SuM[MS] projecting neurons in the SuM[MS]-GtACR1 mice, which expressed the inhibitory protein *Guillardia theta* anion conducting channelrhodopsin 1 (GtACR1)[35] significantly increased the latency to wakefulness from sleep (Supplementary Fig. 10). These results thus indicated that acute inhibition of SuM[MS] projecting neurons reduces wakefulness and increases sleep.

## Discussion

The modulation of wakefulness requires multiple brain regions that span across the entire neural networks[36–38]. Additionally, the identification of wake-active neurons is a necessary step in resolving the mechanism(s) underlying the regulation of wakefulness. For example, monoaminergic neurons in the ascending activating system[38–41] and orexin neurons in lateral hypothalamus[42,43] have been identified as wake-active neurons that are important for the modulation of wakefulness. Here, using optical fiber and optrode recordings, we found a group of wake-active MS projecting neurons in SuM (Fig. 1 and Fig. 2). Optogenetic stimulation of axonal terminals from these SuM[MS]

projecting neurons was sufficient to induce a rapid and reliable transition to wakefulness from sleep (Fig. 3). Retrograde projection-specific labeling combined with chemogenetic manipulation revealed that the SuM[MS] neurons play an important role in promoting wakefulness (Fig. 4 and Fig. 5).

Previous work has shown that MS glutamatergic neurons are all wake-active and involved in wakefulness control[18], and SuM neurons can activate hippocampal neurons during REM sleep and locomotion[8,9,31]. In addition, our previous study revealed a REM-active pattern in all SuM-hippocampus projecting neurons and that these neurons are critical for episodic memory consolidation[4]. However, for SuM[MS] projecting neurons, the firing patterns appear more complicated in different behavioral states. SuM neurons exhibit high activity during exploration and approach behaviors, but low activity during sucrose consumption[19]. Our results show that approximately half (13/23) of the SuM[MS] projecting neurons are wake-active, while about 43% (10/23) neurons are REM-active. While these two groups of neurons exhibit no differences in spike shape and peak firing rate during sleep-wakefulness cycles, wake-active neurons tend to exhibit a higher occurrence of burst firing compared to REM-active neurons (Supplementary Fig. 6). Additionally, following optogenetic activation of SuM[MS] projecting neurons, we observed that inducing wakefulness from REM sleep was more challenging than from NREM sleep (lower success rate and longer latency in Fig. 3d-i). It is possible that these REM-active SuM[MS] projecting neurons might participate in certain REM sleep-related functions, such as memory consolidation[4,15,44] or cortical plasticity[45,46]. Hence, the activation of these REM-active neurons, which are probably recruited for other functions, is insufficient to induce wakefulness promptly and reliably.

To investigate the role of SuM-MS projection in the transition to wakefulness versus its maintenance, we examined the changes in population $Ca^{2+}$ and single-neuron activities of this projection during transitional periods. We observed an increase in activity at the transitions from NREM to wakefulness and from REM to wakefulness, followed by a rapid decrease (Fig. 1f, Fig. 2f, and j). Notably, the wake-active SuM[MS] projecting neurons showed activation several seconds prior to the onset of wakefulness (Fig. 2j), both during the transition from NREM to wakefulness and from REM to wakefulness. These findings indicate that the SuM-MS projection plays a more prominent role in the preparation and initiation of wakefulness rather than its maintenance[10,18].

SuM mainly contains Vgat, Vglut2, Tac1 and Nos1 neurons[2,8,10,19], among which Tac1 neurons project to the septum, hippocampus and other regions, functioning in the control of locomotion[8]. And SuM Nos1/Vglut2 co-expressing neurons contribute to theta rhythm during REM sleep[10]. Immunostaining

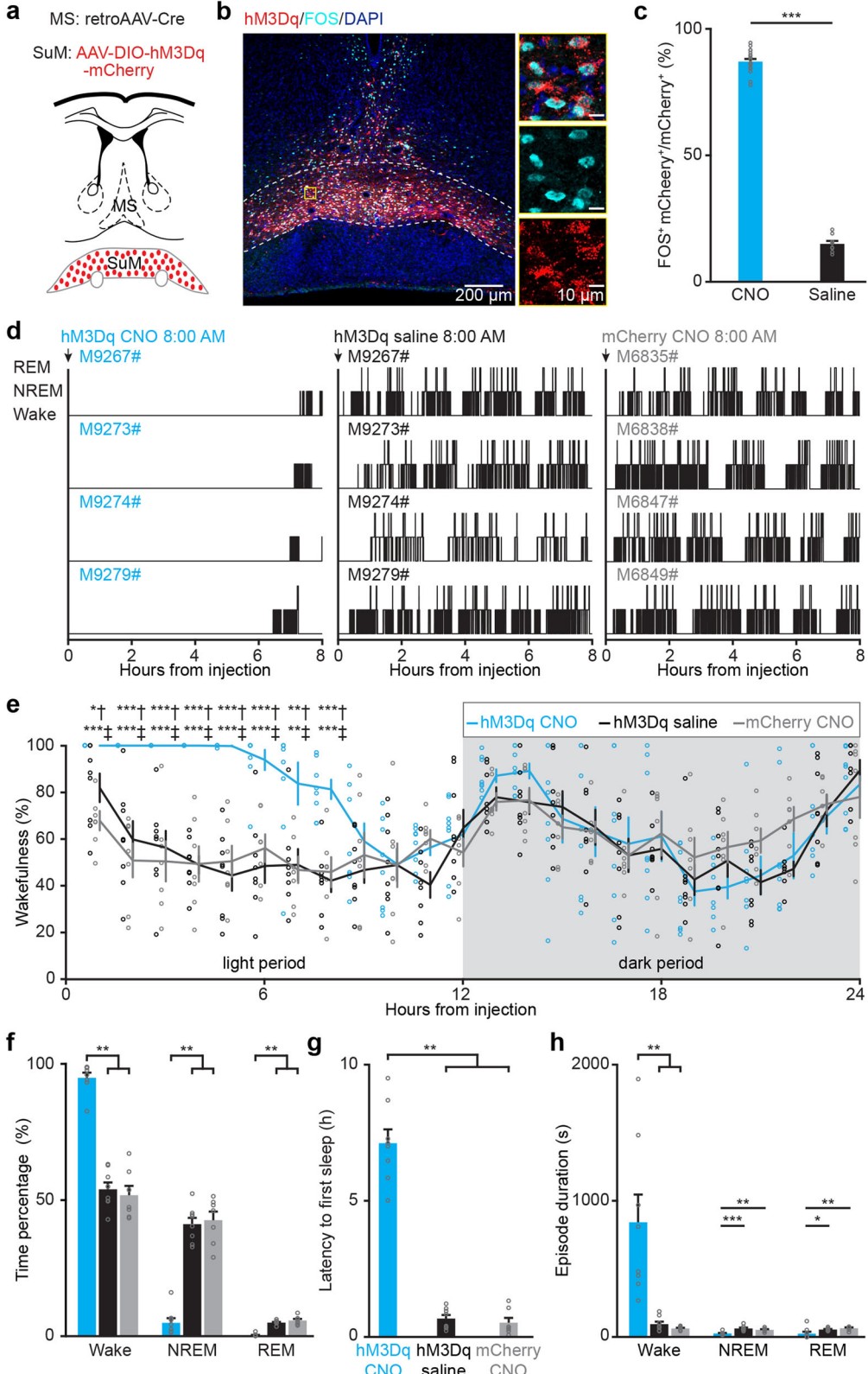

for Nos1 revealed that approximately 17.6% of SuM[MS] projecting neurons are Nos1-containing neurons (Supplementary Fig. 3). However, this group of Nos1 neurons does not appear to contribute significantly to theta activity, as chemogenetic inhibition did not affect EEG activity during wakefulness, NREM sleep, or REM sleep. These results suggest that the MS-projecting Nos1-containing SuM neurons are less involved in theta activity.

In order to determine the connection type of SuM-MS projection, we conducted patch-clamp electrophysiological recordings in brain slice preparations. Our findings revealed that the SuM-MS projection exhibited a monosynaptic excitatory connection (Supplementary Fig. 2). This observation is consistent with previous studies that have also demonstrated the excitatory nature of the SuM-MS projection[1,19,47]. And SuM Vglut2 neurons

**Fig. 4 Chemogenetic activation of SuM^MS projecting neurons increases wakefulness. a** Schematic for virus injection of retroAAV-Cre into MS and AAV-DIO-hM3Dq-mCherry into SuM. **b** Post-hoc histological image of hM3Dq expression in SuM and c-Fos expression after intraperitoneal injection of CNO; yellow rectangle indicates magnified area. **c** Percentage of c-Fos+ mCherry+ neurons among the total mCherry+ neurons in saline- and CNO-treated mice. Saline: $n = 9$ brain sections from 3 mice, CNO: $n = 18$ brain sections from 6 mice; Wilcoxon rank-sum test, ***$p < 0.001$. **d** 8-h hypnograms of hM3Dq-mCherry mice injected with CNO (left), saline (middle), or mCherry mice injected with CNO (right). **e** Hourly percentage of time in wakefulness across a 24-hour recording period after saline injection, CNO injection in hM3Dq-mCherry, or CNO injection in mCherry mice; hM3Dq CNO group: $n = 8$ mice, hM3Dq saline group: $n = 8$ mice, mCherry CNO group: $n = 7$ mice; RMs 2-way ANOVA with Sidak post-hoc comparison test. **f** Percentage of time in each state during the first 8 h after saline or CNO injection; hM3Dq-mCherry group, $n = 8$ mice; mCherry group, $n = 7$ mice; Kruskal-Wallis 1-way ANOVA with Tukey post hoc comparison, **$p < 0.01$. **g** Summary of the latency to first sleep after saline or CNO injection; hM3Dq CNO group: $n = 8$ mice, hM3Dq saline group: $n = 8$ mice, mCherry CNO group: $n = 7$ mice; Kruskal-Wallis 1-way ANOVA with Tukey post hoc comparison, **$p < 0.01$. **h** Episode duration of wakefulness, NREM sleep, and REM sleep during the 8 h period after CNO or saline injection; hM3Dq CNO group: $n = 8$ mice, hM3Dq saline group: $n = 8$ mice, mCherry CNO group: $n = 7$ mice; 1-way ANOVA with LSD post hoc comparison, *$p < 0.05$, **$p < 0.01$, ***$p < 0.001$. Data are represented as mean ± SEM.

can monosynaptically innervate MS Vglut2 neurons by releasing glutamate[13,19]. Thus, wake-active SuM^MS projecting neurons identified here are likely to be glutamatergic and innervate MS Vglut2 neurons, which are known to control wakefulness by activating lateral hypothalamus glutamatergic neurons[18,48]. Furthermore, SuM receives substantial inputs from arousal-related brain regions, including lateral hypothalamus, basal forebrain, locus coeruleus, and dorsal raphe[2,5,10,40,49]. These inputs may contribute to the wakefulness promoting function of SuM-MS projection. Therefore, this SuM-recruited circuit, acts as one part of the whole sleep-wake regulation system, may support locomotion or memory encoding or wakefulness-related behaviors[5,8]. In future studies, it will be important to manipulate wake-active and REM-active neurons within SuM-MS projections separately to further elucidate their individual roles. New experiments, like projection-specific manipulation and electrophysiological recording, can be helpful to study the interaction of these two groups of neurons. In addition, it is worth investigating the possible roles of REM-active SuM^MS projecting neurons in memory consolidation or cortical plasticity. This study was conducted on male mice. To gain a comprehensive understanding of this projection's involvement in wakefulness modulation, further studies involving female animals are also necessary.

## Methods

**Animals**. 12–20-week-old C57BL/6J mice (male) were used in the recording and manipulation experiments. Mice were housed in groups under a constant temperature (21–24°C) and humidity (50%–60%), while those implanted with optical fibers or optrodes were maintained in individual cages. All mice were housed under a 12/12-hour light/dark cycle (with lights on at 8:00 am), and had free access to food and water. All experimental procedures were approved by the Third Military Medical University Animal Care and Use Committee.

**Virus**. AAV2/8-EF1α-eGFP (titer: $1.49 \times 10^{13}$ viral particles/mL) and retroAAV2/2 Plus-EF1α-eGFP (titer: $1.92 \times 10^{13}$ viral particles/mL) were used for tracing experiments. AAV2/9-Syn-axon-jGCaMP7b (titer: $2.17 \times 10^{13}$ viral particles/mL) was used for Ca²⁺ recording. AAV2/9-EF1α-DIO-hChR2-mCherry (titer: $3.67 \times 10^{13}$ viral particles/mL) and retroAAV2/2 Plus-Syn-Cre (titer: $1.92 \times 10^{13}$ viral particles/mL) were used for optrode recording. AAV2/9-syn-hChR2-mCherry (titer: $1.72 \times 10^{13}$ viral particles/mL) was employed for in vitro electrophysiological recording. AAV2/9-CaMKII-hChR2-mCherry (titer: $1.72 \times 10^{13}$ viral particles/mL), AAV2/9-CaMKII-mCherry (titer: $1.72 \times 10^{13}$ viral particles/mL), retroAAV2/2 Plus-Syn-Cre (titer: $1.92 \times 10^{13}$ viral particles/mL), AAV2/9-DIO-hGtACR1-mCherry (titer: $1.72 \times 10^{13}$ viral particles/mL), AAV2/9-DIO-hM3Dq-mCherry

(titer: $1.00 \times 10^{12}$ viral particles/mL), AAV2/9-DIO-hM4Di-mCherry (titer: $1.00 \times 10^{12}$ viral particles/mL), and AAV2/9-DIO-mCherry (titer: $1.00 \times 10^{12}$ viral particles/mL) were used for optogenetic and chemogenetic manipulations. All of the AAV constructs mentioned above were purchased from Taitool Bioscience Co., Ltd. (Shanghai, China) or Obio Biotechnology Co., Ltd. (Shanghai, China).

**Optrode construction for in vivo recording**. An optrode was constructed from a tetrode, a microdrive, and the optics part[50]. A tetrode was grouped by four insulated tungsten wires (25 μm diameter, California Fine Wire). The four tetrodes were arranged into a line with a spacing of ~200 μm and fixed by a fused silica capillary tube, and then mounted onto a micro-drive for vertical movement. The optical fiber (200 μm diameter, NA 0.37) was fixed to tetrodes with the tips being ~500 μm shorter than the tetrode tips. The light from a laser diode (450 nm) was collimated to the optical fiber at the opposite end, with a maximal light intensity measured by an optical power meter (PM100D, Thorlabs). Optical adhesive was used to connect the laser diode and optical fiber.

**Surgical procedures**. For all surgeries, mice were anesthetized with 3% isoflurane in oxygen for 3–5 min and then placed into a stereotaxic frame with an isoflurane concentration maintained at 1%-2%. A heating pad was put under the mice to maintain a temperature of ~37 °C throughout the surgery process. After surgery, the mice were placed back in warm cages and allowed to fully recover. Moreover, they received one dose of dexamethasone sodium phosphate (1 mg/ml, 0.1 ml/10 g/d) and ceftriaxone sodium (50 mg/ml, 0.1 ml/10 g/d) per day by intraperitoneal injection for 3 consecutive days to reduce inflammation[51,52].

For virus injection, 8–12-week-old mice were used. A glass pipette (tip diameter: 10-20 μm) was inserted through a small craniotomy (0.5 × 0.5 mm) to deliver the virus to specific brain areas with an injection speed of approximately 2 nL/s. To express eGFP in the SuM-MS projection, ~50 nL of AAV-eGFP was injected into SuM (AP: -2.8 mm, ML: 1.0 mm, 5° angle toward the midline, DV: 5.0 mm) or ~200 nL of retro AAV2/2-eGFP was injected into MS (AP: 1.0 mm, ML: 0.5 mm, 5° angle toward the midline, DV: 3.8 mm). To express ChR2 or mCherry in the axons of SuM-MS projection, ~200 nL of AAV-hChR2-mCherry or AAV-mCherry was injected into SuM. To express ChR2, hM3Dq, hM4Di, hGtACR1, or mCherry specifically in SuM^MS projecting neurons, ~200 nL of retroAAV2/2-Cre was injected into MS concurrent with ~200 nL of AAV-DIO-ChR2-mCherry, AAV-DIO-hM3Dq-mCherry, AAV-DIO-hM4Di-mCherry, AAV-DIO-hGtACR1-mCherry, or AAV-DIO-mCherry was injected into

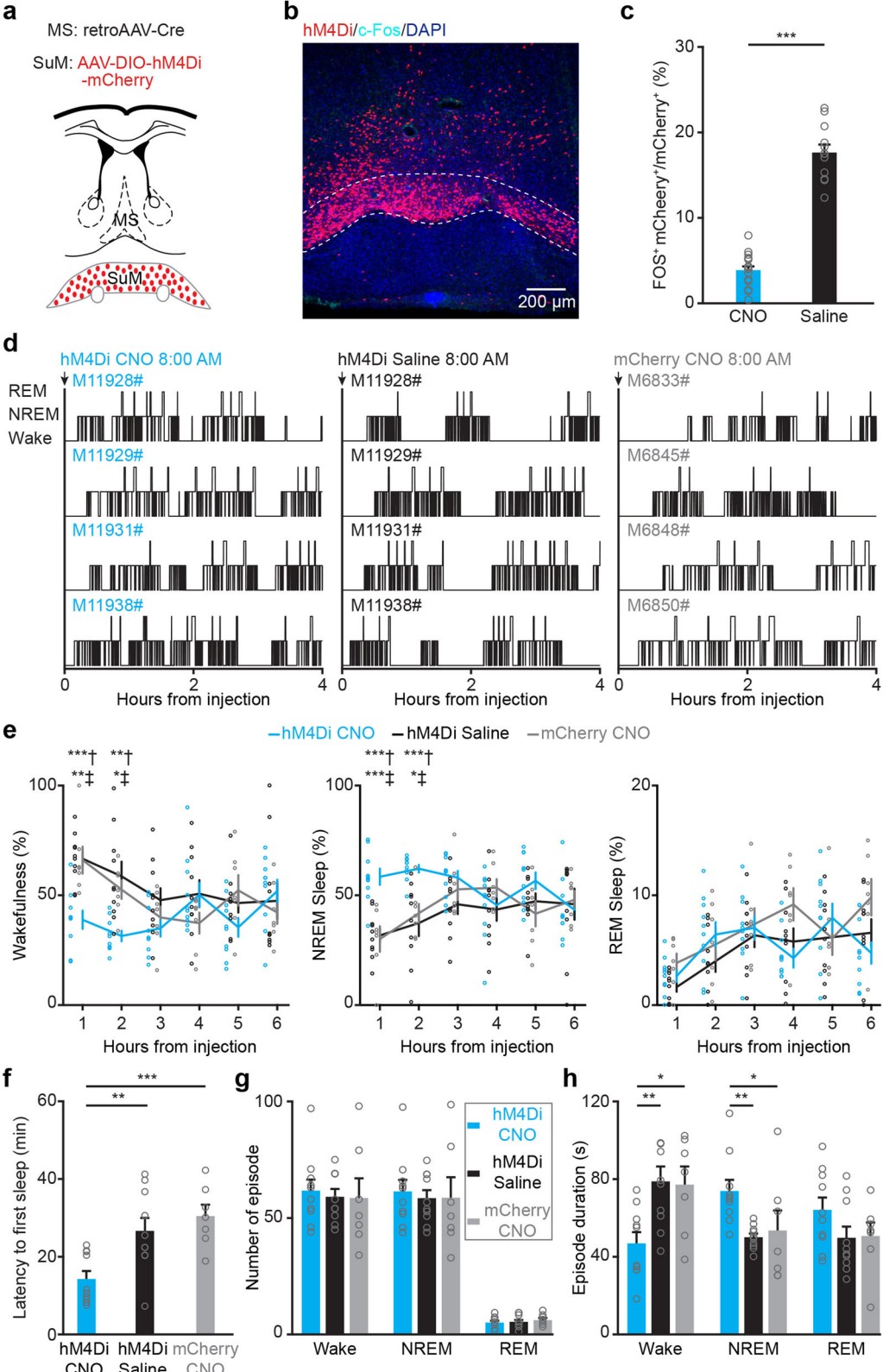

SuM. The viruses were allowed sufficient expression for about one month before subsequent experiments.

For fiber implantation, mice injected with AAV-axon-jGCaMP7b, AAV-CaMKII-ChR2-mCherry, AAV-DIO-hGtACR1-mCherry, or AAV-CaMKII-mCherry were used. To record Ca²⁺ activity, a self-made fiber probe[24] was prepared with an optical fiber (200 μm diameter, NA 0.53, MFP_200/230/900-0.53, Doric lenses) glued into

a mental cannula (ID:0.51 mm, OD: 0.82 mm) after the end face was cut flat. To deliver blue light for optogenetic excitation at axonal terminals in MS, optical fiber ferrules (200 μm diameter, NA 0.37, 5 mm length, Hangzhou Newdoon Technology Co., Ltd) were used. The prepared fiber probe was inserted through a small cranial window above MS (AP: 1.0 mm, ML: 0.5 mm, 5° angle toward the midline) to a depth of 3.5 mm. To deliver blue light for optogenetic

**Fig. 5 Chemogenetic inhibition of SuM$^{MS}$ projecting neurons reduces wakefulness. a** Schematic for injection of retroAAV-Cre into MS and AAV-DIO-hM4Di-mCherry into SuM. **b** Post-hoc histological image of hM4Di expression in SuM and c-Fos expression after intraperitoneal injection of CNO. **c** The percentage of c-Fos$^+$ mCherry$^+$ neurons among all mCherry$^+$ neurons of saline- and CNO-treated mice. Saline: $n = 12$ brain sections from 3 mice, CNO: $n = 18$ brain sections from 5 mice; unpaired $t$-test, ***$p < 0.001$. **d** 4-h hypnograms of hM4Di-mCherry mice injected with CNO (left), saline (middle), or mCherry mice injected with CNO (right). **e** Hourly percentage of time in wakefulness (left), NREM sleep (middle), and REM sleep (right) during the 6 hours following saline or CNO injection in hM4Di-mCherry mice, or CNO injection in mCherry mice; hM4Di CNO: $n = 10$ mice, hM4Di saline: $n = 10$ mice, mCherry CNO: $n = 7$ mice; RMs 2-way ANOVA with Sidak post-hoc comparison test, *$p < 0.05$, **$p < 0.01$, ***$p < 0.001$. **f** Summary of the latency to first sleep after saline or CNO injection; hM4Di CNO: $n = 10$ mice, hM4Di saline: $n = 10$ mice, mCherry CNO: $n = 7$ mice; 1-way ANOVA with LSD post hoc comparison, **$p < 0.01$, ***$p < 0.001$. **g** Episode number of wakefulness, NREM sleep, and REM sleep during the 2 h period after CNO or saline injection; hM4Di CNO: $n = 10$ mice, hM4Di saline: $n = 10$ mice, mCherry CNO: $n = 7$ mice; 1-way ANOVA. **h** Episode duration of wakefulness, NREM sleep and REM sleep during 2 h after CNO or saline injection; hM4Di CNO: $n = 10$ mice, hM4Di saline: $n = 10$ mice, mCherry CNO: $n = 7$ mice; 1-way ANOVA with LSD post hoc comparison, *$p < 0.05$, **$p < 0.01$. Data are represented as mean ± SEM.

inhibition to SuM, optical fiber ferrules (200 μm diameter, NA 0.37, 6 mm length, Hangzhou Newdoon Technology Co., Ltd) were inserted toward SuM (AP: –2.8 mm, ML: ± 2 mm, with a 20° angle towards the midline) at a depth of 4.6 mm. Blue light-curing dental cement (595989WW, Tetric EvoFlow) was applied to fix the probe to the skull. Further reinforcement was achieved with a common dental cement mixture in super glue.

For optrode implantation, mice expressing ChR2 in SuM$^{MS}$ projecting neurons were used. Similarly, the previously described optrode was inserted after a craniotomy above SuM was made. The implantation depth was 4.7 mm from the dura. After a full recovery (the body weight started to increase), the optrode was gradually advanced to the target depth of ~5.0 mm by micro-drive.

For EEG-EMG electrodes implantation, three EEG electrodes made by stainless steel screws were inserted into the craniotomy holes, with two above the frontal lobe (AP: 1.3 mm, ML: ± 1.2 mm) and the third one above the parietal lobe (AP: –3.2 mm, ML: 3.0 mm). Two fine-wire EMG electrodes were inserted into the neck musculature for EMG recording.

Before all recording and manipulation experiments, mice were connected to optical and electrophysiological recording cables in the recording cages to habituate for 3 consecutive days.

**Fiber recording**. Fiber photometry system was used for Ca$^{2+}$ recording[23,53]. The recording was performed in jGCaMP7b-expressing mice with a fiber probe implanted in MS. Ca$^{2+}$ activity (2 KHz), EEG-EMG signals (200 Hz), and behavioral videos (25 Hz) were simultaneously recorded across sleep-wakefulness cycles. Offline event makers were used to synchronize these three forms of signals.

**Optrode recording**. Excitation light pulses (450 nm wavelength, 10 ms duration, ~10 mW intensity, 0.5 s interval) were applied in optrode-implanting mice to identify SuM$^{MS}$ projecting neurons. Units evoked by light stimulation with short spike latency (<8 ms for all the units in our data) and high response reliabilities (>73% for all the units in our data) were identified as ChR2-positive neurons. Then electrophysiological recordings (sampled at 20 KHz), EEG-EMG recordings, and behavioral video recordings were simultaneously conducted across sleep-wakefulness cycles in the light phase. After all recordings were finished, an electrical lesion (current with 30 μA intensity and 12 s duration) was made to verify the recording sites.

**Optogenetic activation**. 473 nm blue laser light (MBL-III-473, Changchun New Industries) was delivered through an optical fiber ferrule under the control of a self-written program on the LabVIEW platform (LabVIEW 2014, National Instrument). The intensity of the light was measured with an optical power meter

(PM100D, Thorlabs). For optogenetic activation, the power intensity was adjusted to ~10 mW at the fiber tip. Stimulation pulses with a duration of 10 ms were randomly delivered at frequencies of 1/5/8/10/20 Hz during NREM or REM sleep for 20 s. All optogenetic activation experiments were conducted between 9:00 a.m. and 5:00 p.m.

**Optogenetic inhibition**. In the optogenetic inhibition experiments, the power intensity was adjusted to ~3 mW, and the light was continuously delivered. The EEG and EMG signals were manually monitored by experimenters in real-time, and inhibition was applied after 20 s from the onset of stable NREM or REM sleep until the end of each sleep episode. All optogenetic inhibition experiments were conducted between 9:00 a.m. and 5:00 p.m. The duration of inhibition was 38.5 ± 1.9 s in our dataset (n = 7 mice).

**Chemogenetic manipulation**. Chemogenetic manipulations were applied to hM3Dq or hM4Di-expressing mice after EEG-EMG electrodes implantation. After recovery, CNO (1 mg/kg, dissolved in saline, 0.3 mL) or an equal volume of saline was intraperitoneally injected at 8:00 a.m. EEG-EMG signals and behavioral videos were recorded 2 h before drug applications and lasted for 24 hours.

**In vitro electrophysiological recordings**. Mice injected with AAV-Syn-ChR2-mCherry in SuM were deeply anesthetized with isoflurane. Following decapitation, the brains were carefully removed and immediately submerged into ice-cold oxygenated cutting solution containing (in mM): 110 N-methyl-D-glucamine, 2.5 KCl, 1.2 NaH$_2$PO$_4$, 25 NaHCO$_3$, 25 glucose, 10 MgSO$_4$ and 0.5 CaCl$_2$, adjusted to pH 7.3 with HCl 6 M, 310 mOsm. The brains were sliced into 260 μm-thick coronal sections containing MS with a vibratome (Leica, VT1200S). The brain slices were initially incubated in the oxygenated cutting solution at 35 °C for 5 min. Subsequently, they were transferred to an oxygenated ACSF solution containing (in mM): 119 NaCl, 2.5 KCl, 1.2 NaH$_2$PO$_3$, 25 NaHCO$_3$, 2 MgSO$_4$, 2 CaCl$_2$, 12.5 glucose, adjusted to pH 7.3. The slices were then incubated in the oxygenated ACSF solution at 35 °C for 30 minutes. For recording purposes, the brain slices were transferred to a recording chamber and continuously perfused by oxygenated ACSF throughout the entire recording process.

Whole-cell patch-clamp recordings were performed in the cell bodies of MS neurons. Visualization of MS neurons was achieved using an upright microscope (BX51WI, Olympus) equipped with infrared differential interference contrast (DIC) optics, enabling visualization through transmitted light or epifluorescence. The recordings were conducted with an EPC-10 amplifier (HEKA Elektronik). Recording micropipettes were fabricated from

borosilicate glass capillaries (BF 150-86-10, Sutter Instruments) pulled using a micropipette puller (P-97, Sutter Instruments) with resistances ranging from 4 to 6 MΩ. These micropipettes were filled with an intrapipette solution containing (in mM): 130 cesium methane sulfonate, 5 KCl, 10 HEPES, 2 NaCl, 5 Mg$_2$ATP, 0.4 Na$_2$GTP, 5 EGTA, and 10 disodium phosphocreatine, with a pH of 7.3 and an osmolarity of 300 mOsm. Excitation of ChR2 expressed in SuM axons was achieved by light from a 473-nm laser (MBL-III-473, Changchun New Industries, 10 mW, 2 ms duration) delivered through an optical fiber. EPSCs and IPSCs were recorded at −70 mV and +10 mV respectively. TTX (1 µM, Sigma) was applied to prevent action potential generation, and 4-AP (100 µM, Sigma) was added to block K$_V$1 potassium channels. All signals were digitized at a rate of 20 kHz and low-pass-filtered at 2 kHz using the amplifier circuitry. Whole-cell recordings were excluded if series resistance exceeded 20 MΩ.

**Sleep structure analysis.** All EEG-EMG signals were first band-filtered (EEG: 0.5–30 Hz, EMG: 10–70 Hz). Then signals were divided into non-overlapping epochs of 4 s for analysis. The NREM sleep, REM sleep, or wakefulness state was automatically defined according to the amplitude of EMG and δ/θ power of the EEG spectrum by sleep analysis software (SleepSign for Animal, Kissei Comtec). NREM sleep was characterized by high amplitude in the EEG δ band (0.5–4 Hz) and low amplitude of EMG activity. REM sleep was characterized by low amplitude in the EEG δ band and high amplitude in EEG θ band (4–10 Hz), without tonic EMG activity. Wakefulness was characterized by high EMG activity and low amplitude of EEG activity. The automatically defined results were reviewed and manually corrected. The cumulative duration of NREM sleep, REM sleep, and wakefulness were summarized by a self-written MATLAB program.

**Histology.** All mice used above were perfused with 4% paraformaldehyde (PFA) in PBS. The brains were sectioned into 50-µm slices after being dehydrated with 15% sucrose in 4% PFA for 24 h. Brain sections were imaged by a wide-field fluorescence microscope (Olympus, BX51) or confocal microscope (Zeiss, LSM 700) after being stained with DAPI. For c-Fos immunohistochemistry, mice expressing hM3Dq or hM4Di were perfused 1.5 h after CNO or saline injection and sectioned as described above. Brain sections were blocked and incubated with primary antibodies (rabbit anti-c-Fos 1:200, ab190289, Abcam, RRID: AB_2737414)[54,55]. For Nos1 immunohistochemistry, brain slices containing mCherry-expressing SuM$^{MS}$ projecting neurons were utilized. The slices were incubated with Nos1 primary antibodies (guinea pig anti-Nos1, 1:200, OB-PGP070, Oasis Biofarm, RRID: AB_2940794). The number of c-Fos, Nos1 and mCherry positive neurons was manually counted by experiment-blinded analysts.

**Data analysis.** All Ca$^{2+}$ signals were filtered by a Savitzky-Golay FIR smoothing filter with 50 side points and a 3rd-order polynomial[23,24]. Then Ca$^{2+}$ signals were calculated into Δf/f by the formula of Δf/f = (f - f$_{baseline}$) / f$_{baseline}$, where f$_{baseline}$ represents the baseline fluorescence obtained during recording. To quantify the Ca$^{2+}$ signals during sleep-wakefulness cycles, we identified the arousal state based on synchronous EEG-EMG signals. The area under Ca$^{2+}$ signals was used for statistical analysis[4].

The raw extracellular electrophysiological data were high-pass filtered (250 Hz) to extract the spikes[4,23]. Events that exceeded an amplitude threshold of four standard deviations above the baseline were saved for subsequent spike sorting analysis. All detected events for each tetrode were sorted in the toolbox MClust based on the features of waveforms[56]. The firing rates of each unit were calculated in a sliding time bin of 0.5 s (0.1 s

interval). Units were classified according to the firing rates in NREM sleep, REM sleep, and wakefulness. Burstiness analysis was performed by examining the inter-spike interval (ISI) of each single-neuron. Spike events occurring within a time interval of 3–15 ms ISI were classified as burst events[57]. To quantify burstiness, a bursty score was calculated as the ratio of burst events to the total number of firing events[29,30].

We analyzed the spectral profiles of EEG activity by a self-designed MATLAB program[58]. The EEG data were calculated by fast Fourier transformation with a frequency resolution of 0.15 Hz. The spectral bands were categorized into the following frequency ranges: slow wave (0.1–2 Hz), delta (2–4 Hz), low-theta (4–7 Hz), high-theta (7–12 Hz), alpha (12–15 Hz), and beta (15–30 Hz)[7,10,15]. In the optogenetic activation experiments, the power densities of EEG signals were calculated in the 20-second periods before and during 20 seconds of stimulation at frequencies of 1 Hz, 5 Hz, 8 Hz, 10 Hz, and 20 Hz. These power densities were then compared for analysis.

**Statistics and reproducibility.** Details of the experiments are provided in the main text and in "Materials and methods". Statistical analyses were performed in MATLAB and SPSS22.0 (Supplementary Data 1). Normality tests were analyzed between samples. Parametric tests (paired and unpaired t-tests, RMs 1-way ANOVA with LSD post hoc comparison, 1-way ANOVA with LSD post hoc comparison, and RMs 2-way ANOVA with Sidak's post hoc comparison) were subsequently applied if normality or equal variance was achieved. Otherwise, non-parametric tests (Wilcoxon signed-rank test, Wilcoxon rank-sum test, Kruskal-Wallis test with Tukey post hoc comparison, and Friedman's ANOVA test) were applied. All tests were two-tailed. All summary data were from individual mice and represented as mean ± SEM.

**Reporting summary.** Further information on research design is available in the Nature Portfolio Reporting Summary linked to this article.

### Data availability
All source data underlying the graphs and charts shown in the figure have been uploaded as Supplementary Data 2. All other data are available from the corresponding author (or other sources, as applicable) on reasonable request.

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

## Acknowledgements

The authors are grateful to Ms. Jia Lou for help in composing and layout editing of the figures. This work was supported by grants from the National Natural Science Foundation of China (32200838, 31925018, 32127801, 31921003, 81971236) and the National Key R&D Program of China (2021YFA0805000). X.C. is a member of the CAS Center for Excellence in Brain Science and Intelligence Technology.

## Author contributions

Conceptualization and methodology, H.Q., L.W., and X.C.; software programming, W.J. and X.L.; data curation, M.L., T.J., J.T. and H.Q.; investigation, M.L., T.J., J.T., X.W., Q.C., X.Y., J.Y. and Z.Y.; technical support, R.W., Z.Z. and J.X.; writing-original draft, M.L.; writing – review & editing, H.Q. and X.C.; funding acquisition, L.W., X.C. and H.Q.; resources, M.L., T.J., W.J. and H.Q.; supervision, H.Q. and X.C. All authors read and commented on the manuscript.

## Competing interests

The authors declare no competing interests.
