## [Peer Review File · Communications Biology]

Reviewers' comments:

Reviewer #1 (Remarks to the Author):

Review of Manuscript:

Title: Hypothalamic supramammillary neurons that project to the medial septum control wakefulness

Summary: Liang et al. report the following key observations:

1. The supramammillary region (SuM) neurons projecting to the medial septum (MS)(SuM-to-MS neurons) increased activities more during wakefulness and REM sleep than NREM sleep, using fiber-photometry Ca²⁺-imaging.
2. These neurons were classified into two types: wake-active neurons and REM-sleep active neurons.
 - a. Wake-active neurons: Those increased firing rates at the transition from NREM/REM sleep to wakefulness and decreased firing rates at the transition from wakefulness to NREM sleep.
 - b. REM-sleep active neurons: The other increased firing rates at the NREM-to-REM sleep transition and decreased at the REM sleep-to-wakefulness transition.
3. Optogenetic activation of both types of SuM-to-MS neurons induced wakefulness transitioned from the NREM or REM state in a frequency dependent manner.
4. Similarly, chemogenetic activation of these neurons during the light phase prolonged wakefulness and shortened NREM and REM sleep.
5. Conversely, chemogenetic inactivation of these neurons increased NREM sleep and reduced wakefulness time.

Overall Evaluation: The study provides valuable data for understanding SuM neuron functions, but improvements are needed for more comprehensive understanding: 1. Clarify questionable data analyses (see below); 2. Provide nuanced interpretation (see below).

Specific Comments:

Lines 36-38: The authors concluded "Therefore, these results identify the MS as a functional downstream target of SuM and provide evidence for a causal role for this hypothalamic-septal projection in wakefulness control". This conclusion is too strong given the results. If the inhibition of SuM neurons prevents animals from transitioning from sleep to wakefulness, I support that the conclusion. But the present study did not provide such evidence. Therefore, it is more adequate to characterize that SuM neuron activity modulates wakefulness. Similarly, it should be avoided to conclude that SuM-to-MS neurons play an essential role in promoting wakefulness (lines 235-237), as it is unclear how essential those neurons are.

Lines 79-81: Although the authors state "the specific cell types of SuM neurons projecting to MS . . . have not been well characterized", there is previous work that has investigated the issue as much as the present study did. Kesner et al. (Nat Commun 2021, 12, 2811) showed that SuM glutamatergic neurons project to the MS and excite MS neurons in a glutamate receptor dependent manner. Therefore, this statement is a mischaracterization of the previous work.

Lines 211-216: Please confirm if CNO was administered at 8 am, the onset of the light phase.

Lines 220-224: Clarify the methodology used for photostimulation to inhibit SuM-to-MS neurons. Describe the duration and timing of stimulation in relation to the light-dark cycle.

Line 316: Clarify the use of Wilcoxon signed-rank tests for Fig. 2i. This test is inappropriate for comparing three groups and doesn't account for multiple comparisons.

Lines 339-342: The use of paired t-tests for Fig. 3j and 3k is inappropriate due to multiple comparisons. Consider using one-way ANOVAs, as there are six groups in each EEG frequency analysis.

AAV-retro-Cre: Consistency is essential; either use "AAVRetro-cre" (Figures 2 and 4) or "retroAAV-cre" (Figure 5) consistently throughout the manuscript.

Reviewer #2 (Remarks to the Author):

Liang et al reported that approximately half of the SuM neurons projecting to the medial septum (MS) exhibit wake-promoting activity. Optogenetic stimulation of the axonal terminals originating from SuM-MS projection elicits a prompt and dependable transition from non-rapid-eye movement (NREM) or rapid eye movement (REM) sleep to wakefulness. Furthermore, chemogenetic activation of SuM-MS projecting neurons significantly augments the overall duration of wakefulness and extends the latency period preceding sleep onset. Correspondingly, chemogenetic inhibition of these neurons leads to a substantial reduction in wakefulness duration and diminishes the latency to sleep onset. The author took several advanced technology to prove the SuM-MS projection play a causal role for wakefulness control. The data are very solid. I just have few minor questions:

1. The author found that there are both wakefulness-related neurons and REM-related neurons in SuM. In this study, the author mainly focused on the neural correlates of arousal. Have the authors paid attention to REM-related neurons? How might these two types of neurons interact with each other? No more experiment is required, but the author should discuss it in detail.
2. The authors found that the projection of SuM-MS controls arousal, but did not discuss the specific role of this projection in the whole sleep-wake regulation system. Why does the regulation of wake need this projection?

We would like to express our deep appreciation to the reviewers for their constructive comments on our manuscript (COMMSBIO-23-3306). In order to address each of their comments, we made the following changes. We have marked the major changes in the manuscript in red.

- (1) We have changed “wakefulness control” into “wakefulness modulation”.
- (2) We have corrected the onset of 12/12-hour light/dark cycle.
- (3) We have provided detailed methodology for optogenetic inhibition experiments.
- (4) We have changed the statistical methods in Fig.2i, Fig. 3j and Fig. 3k.
- (5) We have used “retroAAV-Cre” consistently throughout the manuscript.
- (6) We have discussed the REM-active SuM^{MS} projecting neurons in the discussion part.

Reviewer Comments:

Reviewer #1:

Summary: Liang et al. report the following key observations:

1. The supramammillary region (SuM) neurons projecting to the medial septum (MS)(SuM-to-MS neurons) increased activities more during wakefulness and REM sleep than NREM sleep, using fiber-photometry Ca²⁺-imaging.
2. These neurons were classified into two types: wake-active neurons and REM-sleep active neurons.
 - a. Wake-active neurons: Those increased firing rates at the transition from NREM/REM sleep to wakefulness and decreased firing rates at the transition from wakefulness to NREM sleep.
 - b. REM-sleep active neurons: The other increased firing rates at the NREM-to-REM sleep transition and decreased at the REM sleep-to-wakefulness transition.
3. Optogenetic activation of both types of SuM-to-MS neurons induced wakefulness transitioned from the NREM or REM state in a frequency dependent manner.
4. Similarly, chemogenetic activation of these neurons during the light phase prolonged wakefulness and shortened NREM and REM sleep.
5. Conversely, chemogenetic inactivation of these neurons increased NREM sleep and reduced wakefulness time.

Overall Evaluation: The study provides valuable data for understanding SuM neuron functions, but improvements are needed for more comprehensive understanding: 1. Clarify questionable data analyses (see below); 2. Provide nuanced interpretation (see below).

We thank the reviewer very much for the appreciation of our work and the constructive comments that are essential for improving our manuscript.

Specific Comments:

Lines 36-38: The authors concluded “Therefore, these results identify the MS as a functional downstream target of SuM and provide evidence for a causal role for this hypothalamic-septal projection in wakefulness control”. This conclusion is too strong given the results. If the inhibition of SuM neurons prevents animals from transitioning from sleep to wakefulness, I support that the conclusion. But the present study did not provide such evidence. Therefore, it is more adequate to characterize that SuM neuron activity modulates wakefulness. Similarly, it should be avoided to conclude that SuM-to-MS neurons play an essential role in promoting wakefulness (lines 235-237), as it is unclear how essential those neurons are.

Thanks a lot for this suggestion. We now tune down this statement. Following the reviewer’s

suggestion, we changed “provide evidence for a causal role for this hypothalamic-septal projection in wakefulness control” to “provide evidence for the modulation of wakefulness by this hypothalamic-septal projection”. In addition, we changed “SuM-to-MS neurons play an essential role in promoting wakefulness” to “SuM-to-MS neurons play an important role in promoting wakefulness”. We avoided the use of “essential”.

Lines 79-81: Although the authors state “the specific cell types of SuM neurons projecting to MS . . . have not been well characterized”, there is previous work that has investigated the issue as much as the present study did. Kesner et al. (Nat Commun 2021, 12, 2811) showed that SuM glutamatergic neurons project to the MS and excite MS neurons in a glutamate receptor dependent manner. Therefore, this statement is a mischaracterization of the previous work.

Following the reviewer’s suggestion, we have removed this statement.

Lines 211-216: Please confirm if CNO was administered at 8 am, the onset of the light phase.

Yes, the CNO was administered at 8:00 am. And the 12/12-hour light/dark cycle in Materials and methods part was written into 7:00 am by mistake. It should be 8:00 am. Now we have corrected it.

Lines 220-224: Clarify the methodology used for photostimulation to inhibit SuM-to-MS neurons. Describe the duration and timing of stimulation in relation to the light-dark cycle.

We have now clarified the methodology used for optogenetic inhibition of SuM^{MS} projecting neurons in materials and methods part. Optogenetic inhibition was applied after 20 s from the onset of stable NREM or REM sleep until the end of each sleep episode. All optogenetic inhibition experiments were conducted in the light phase between 9:00 a.m. and 5:00 p.m. The duration of inhibition was 38.5 ± 1.9 s in our dataset.

Line 316: Clarify the use of Wilcoxon signed-rank tests for Fig. 2i. This test is inappropriate for comparing three groups and doesn't account for multiple comparisons.

We have now changed the statistical method into Friedman's ANOVA test with post hoc comparison in Fig. 2i.

Lines 339-342: The use of paired t-tests for Fig. 3j and 3k is inappropriate due to multiple comparisons. Consider using one-way ANOVAs, as there are six groups in each EEG frequency analysis.

We have now changed the statistical method into RM one-way ANOVA test with LSD post-hoc comparison in Fig. 3j and 3k.

AAV-retro-Cre: Consistency is essential; either use "AAVRetro-cre" (Figures 2 and 4) or "retroAAV-cre" (Figure 5) consistently throughout the manuscript.

We have changed them into “retroAAV-Cre” throughout the manuscript.

Reviewer #2:

Liang et al reported that approximately half of the SuM neurons projecting to the medial septum (MS) exhibit wake-promoting activity. Optogenetic stimulation of the axonal terminals originating

from SuM-MS projection elicits a prompt and dependable transition from non-rapid-eye movement (NREM) or rapid eye movement (REM) sleep to wakefulness. Furthermore, chemogenetic activation of SuM-MS projecting neurons significantly augments the overall duration of wakefulness and extends the latency period preceding sleep onset. Correspondingly, chemogenetic inhibition of these neurons leads to a substantial reduction in wakefulness duration and diminishes the latency to sleep onset. The author took several advanced technology to prove the SuM-MS projection play a causal role for wakefulness control. The data are very solid. I just have few minor questions:

We thank the reviewer very much for the appreciation of our work and the constructive comments that are essential for improving our manuscript.

1. The author found that there are both wakefulness-related neurons and REM-related neurons in SuM. In this study, the author mainly focused on the neural correlates of arousal. Have the authors paid attention to REM-related neurons? How might these two types of neurons interact with each other? No more experiment is required, but the author should discuss it in detail.

Thanks a lot for this important suggestion. It is possible that these REM-active SuM^{MS} projecting neurons might participate in certain REM sleep-related functions, such as memory consolidation or cortical plasticity. New experiments, such as projection-specific manipulation and electrophysiological recording, are needed to investigate the interaction of wake-active and REM-active SuM^{MS} projecting neurons in future. We have discussed it in the discussion part.

2. The authors found that the projection of SuM-MS controls arousal, but did not discuss the specific role of this projection in the whole sleep-wake regulation system. Why does the regulation of wake need this projection?

We now included the discussion of the possible circuit mechanisms of SuM-MS projection in arousal control. SuM receives inputs from arousal-related brain regions, including lateral hypothalamus, basal forebrain, locus coeruleus, and dorsal raphe. These arousal-related brain regions may active SuM-MS projection, and then active MS Vglut2 neurons by releasing glutamate. And the MS Vglut2 neurons have been reported to control wakefulness by activating lateral hypothalamus glutamatergic neurons (An et al., 2021). Therefore, this SuM-recruited circuit, acts as one part of the whole sleep-wake regulation system, may support locomotion or memory encoding or wakefulness-related behaviors (Farrell et al., 2021; Chen et al., 2020).

REVIEWERS' COMMENTS:

Reviewer #1 (Remarks to the Author):

The authors have fully addressed all my concerns.

Reviewer #2 (Remarks to the Author):

The author already addressed my questions.